# Numerical Investigations on Scour and Flow around Two Crossing Pipelines on a Sandy Seabed

**Fan Zhang** [1] , **Zhipeng Zang** [1,2,*] , **Ming Zhao** [3] , **Jinfeng Zhang** [1,2] , **Botao Xie** [4] and **Xing Zou** [4]

1   State Key Laboratory of Hydraulic Engineering Simulation and Safety, Tianjin University, Tianjin 300350, China
2   Key Laboratory of Earthquake Engineering Simulation and Seismic Resilience of China Earthquake Administration, Tianjin University, Tianjin 300350, China
3   School of Engineering, Design and Built Environment, University of Western Sydney, Penrith 2751, Australia
4   CNOOC Research Institute Co., Ltd., Beijing 100028, China
*   Correspondence: zhipeng.zang@tju.edu.cn

**Abstract:** When a pipeline is laid on the seabed, local scour often occurs below it due to sea currents. In practical engineering, there are some cases that two pipelines laid on the seabed need to cross with each other. The complex flow structures around two crossing pipelines make the scour characteristics different from that of an isolated single pipeline. In this study, scour below two crossing pipelines was simulated numerically using the CFD software Flow-3D. The study is focused on the effect of the intersecting angle on the equilibrium depth and time scale of scour below the crossing position. Five intersecting angles, i.e., $\alpha = 0°$, $15°$, $30°$, $45°$ and $90°$, are considered, where $\alpha = 0°$ and $90°$ represent two pipelines parallel and perpendicular to each other, respectively. The results show that the equilibrium depth and the time scale of scour below the two crossing pipelines are greater than those of an isolated single pipeline. The equilibrium depth and time scale of scour have the largest values at $\alpha = 0°$ and decrease with the increase of the intersecting angle. Finally, the flow structures around the crossing pipelines are presented to explain the scour process.

**Keywords:** crossing pipelines; local scour; flow structure; intersecting angle; numerical simulation

## 1. Introduction

Submarine pipelines are the main form of offshore oil and gas transportation because of their advantages of stable transportation, low cost and weak influence by climate. However, the flow structures around the submarine pipeline are complex, and the local scour often occurs below the pipeline due to the local amplification of the flow velocity. Local scour has a great impact on the operation safety of the submarine pipeline as it forms free spans and increases the risk of bending failure due to gravity and fatigue failure due to vortex-induced vibrations [1]. The survey on submarine pipeline accidents shows that local scour and the resulting free span is one of the main reasons for pipeline failure. With the rapid development of the offshore oil and gas industry, there are many cases that two pipelines cross with each other. The flow fields and the sediment transport around these pipelines are more complex, and the local scour is more severe than a single pipeline. Thus, it is of great engineering and academic significance to study the local scour around the crossing pipelines.

Two-dimensional (2D) scour around a submarine pipeline has been studied extensively. In 2D scour, the flows and local scour are assumed to be uniform along the pipeline span. Mao [2] experimentally studied the relationship between the equilibrium scour depth of a pipeline and the nondimensional bed shear stress under currents. Fredsøe et al. [3] proposed a useful empirical formula for predicting the time scale for scour below a pipeline under the live-bed condition. Sumer et al. [4] found that the piping phenomenon is the dominant cause of the initiation of local scour. Gao et al. [5] experimentally investigated the

coupling effects between pipeline vibration and sand scour. Yang et al. [6] studied the scour depth for a submarine pipeline with a spoiler. Mohr et al. [7] found that the scour hole shape and the rate of scour are highly affected by the erosion properties of the sediment, and two new empirical formulae were put forward to predict the time scale of scour process. Zang et al. [8] studied the scour depth of a pipeline in different flow incident angles and proposed relevant empirical formulae. In addition to laboratory tests, various numerical models have also been developed to study the local scour around pipelines. Brørs [9] established a 2D numerical model, considering suspended and bedload sediment transport, and the predicted 2D bed profiles were in good agreement with experimental results. Liang et al. [10] found that the standard *k-ε* model was preferable to the Smagorinsky subgrid-scale (SGS) model in the 2D scour simulation. Zhao and Cheng [11] conducted a numerical investigation on local scour below a vibrating pipeline under steady currents. Zhao and Cheng [12] numerically simulated the local scour below piggyback pipelines under currents. Zang et al. [13] numerically studied the mechanism of the onset of scour below a pipeline in currents/waves. Fuhrman et al. [14] conducted a numerical simulation of wave-induced scour and backfilling processes beneath a submarine pipeline. Liu et al. [15] conducted numerical modeling of local scour and forces for a submarine pipeline under surface waves.

Practically, scour below a pipeline occurs in a three-dimensional (3D) way, i.e., the scour initiates at a weak point and then propagates in two opposite directions along the pipeline, forming a free span. Cheng et al. [16,17] conducted physical experiments and proposed empirical formulae for calculating the spanwise propagation speed of 3D scour below a pipeline under currents, waves and combined currents and waves. Wu and Chiew [18] found that the propagation speed of the 3D scour below a pipeline was very sensitive to the Froude number in addition to the Shields parameter. Wu et al. [19] suggested that the differential pressure and the local bed shear stress were important factors affecting the pipeline scour propagation in the spanwise direction. Sui et al. [20] proposed a new universal model for predicting the span migration rate of 3D scour below a pipeline in both clear-water and live-bed regimes.

Through 3D numerical studies, Alam and Cheng [21] confirmed that the vortex shedding played an important role in the generation of lee-wake scour and found that the scour slope at the shoulder region hardly changed during the whole scour process. Cheng and Zhao [22] found that the high Shields parameter and the steep bed slopes at the two shoulders contributed to the scour propagation in the spanwise direction. Xu [23] established a 3D scour model based on Flow-3D software to predict the development of the scour hole below the pipeline and studied the flow structures. Shi et al. [24] studied the 3D local scour below a pipeline based on the Ansys-Fluent software through Users-Defined-Functions (UDFs).

At present, the research on 3D scour of pipelines is mainly aimed at a single pipeline or two parallel pipelines. There are no published studies regarding flow and scour around two crossing pipelines with an angle between them [20,24]. In this paper, the Flow-3D software is used to study the 3D scour problem of two crossing pipelines. The full paper is divided into five sections: Section 1 introduces the research background and significance of the work; Section 2 describes the governing equations, the establishment and validations of the model; Section 3 carries out a numerical simulation to predict the equilibrium depth and time scale of scour below the two crossing pipelines; Section 4 analyzes the flow structures around the two pipelines and the mechanism of scour development; several concluding remarks are given in the final section.

## 2. Model Setup and Validation

### 2.1. Governing Equations

In the present study, the water flow is simulated by solving the Reynolds-Averaged Navier–Stokes (RANS) equations of incompressible fluid. The two-equation *k-ε* turbulence model is used to close the equations. The free surface is captured by using VOF technology.

The RANS equations and $k$-$\varepsilon$ turbulence equations are solved by the software Flow-3D. For the details of the flow simulation, one can refer to the Flow-3D user manual [25].

The scour process is predicted by the embedded functions of Flow-3D for sediment transport, which includes both bedload and suspended load. The suspended load transport is calculated by solving the convection–diffusion equations of the suspended sediment concentration.

The bedload transport is calculated with the empirical formula by Van Rijn [26]. The bedload transport rate $u_{\text{bedload}}$ is related to the volumetric transport rate $q_b$

$$u_{\text{bedload}} = \frac{q_b}{\delta f_b} \tag{1}$$

where $f_b$ is the critical packing fraction of the sediment, taken as 0.64 for spheric sand particles; $\delta$ is the bedload layer thickness. The volumetric transport rate $q_b$ is obtained by:

$$q_b = \Phi g \left( \frac{\rho_s - \rho_f}{\rho_f} d_{50}^3 \right)^{0.5} \tag{2}$$

where g is the acceleration due to gravity; $\rho_s$ and $\rho_f$ are the density of sand and water, respectively; $d_{50}$ is the median grain size of sand; $\Phi$ is the nondimensional bedload transport rate and can be calculated with the empirical formula by Van Rijn [26]:

$$\Phi = \beta_{\text{VR}} d_*^{-0.3} \left( \frac{\theta}{\theta_{cr}'} - 1.0 \right)^{2.1} c_b \tag{3}$$

where $\beta_{\text{VR}}$ is bedload transport coefficient, taken as 0.053; $c_b$ is the volume fraction of the sand bed; $d_*$ is the nondimensional particle size and defined as:

$$d_* = d_{50} \left[ \frac{\rho_f \left( \rho_s - \rho_f \right) g}{\nu^2} \right]^{1/3} \tag{4}$$

where $\nu$ is kinematic viscosity.

The bedload layer thickness $\delta$ is calculated by:

$$\frac{\delta}{d_{50}} = 0.3 d_*^{0.7} \left( \frac{\theta}{\theta_{cr}'} - 1 \right)^{0.5} \tag{5}$$

In Equations (3) and (5), $\theta$ is the Shields parameter, and its critical value for sediment incipient motion is denoted as $\theta_{cr}$. $\theta_{cr}'$ is the modified critical Shields parameter for slope effect. Here, the critical Shields parameter on a flat bed is calculated by the Soulsby–Whitehouse Equation [27]:

$$\theta_{cr} = \frac{0.3}{1 + 1.2 d_*} + 0.055[1 - \exp(-0.02 d_*)] \tag{6}$$

Then, the modified critical Shields parameter is expressed as follows:

$$\theta_{cr}' = \theta_{cr} \frac{\cos \psi \sin \beta + \sqrt{\cos^2 \beta \tan^2 \varphi - \sin^2 \psi \sin^2 \beta}}{\tan \varphi} \tag{7}$$

where $\beta$ is the angle of slope of bed, $\varphi$ is the angle of repose for sediment particles (default at 32°) and $\psi$ is the angle between the flow and the upslope direction.

*2.2. Model Validation*

2.2.1. 2D Scour Simulation and Mesh Sensitivity Analysis

A test case of 2D scour below a pipeline conducted in a flume by Mao [2] is simulated. The sediment parameters in the numerical model are the same as those used in the experiment listed in Table 1. In the test case, the Shields parameter in a far field is 0.040 for a current velocity of 0.35 m/s, while the critical value for the sand is 0.048, which indicates that the scour is in a clear-water regime.

**Table 1.** 2D single-pipeline scour model parameters.

| Inlet Velocity $v_o$ (m/s) | Pipeline Diameter $D$ (m) | Sediment Density $\rho$ (kg/m$^3$) | Medium Diameter $d_{50}$ (mm) | Embedment Ratio $e/D$ |
|---|---|---|---|---|
| 0.35 | 0.1 | 2650 | 0.36 | 0 |

The 2D computational domain established in the *y-z* coordinate system is shown in Figure 1. The water flume is 3.0 m in length and 0.7 m in height with a 0.2 m thick sand bed. The still water depth is 0.35 m. A submarine pipeline with a diameter of $D$ = 0.1 m is laid on the sandy seabed without a gap, i.e., the initial embedment depth $e$ = 0. The numerical model setup is shown in Figure 1. The direction of water flow is in the positive *y*-direction. The length of the sand bed upstream of the pipeline is 20 $D$, which can guarantee a fully developed current profile at the pipeline. An initial scour hole of 0.1 $D$ is set below the pipeline to simulate the onset of scour. Previous studies have shown that the initial scour hole will have little effect on the scour process and the final scour depth [16].

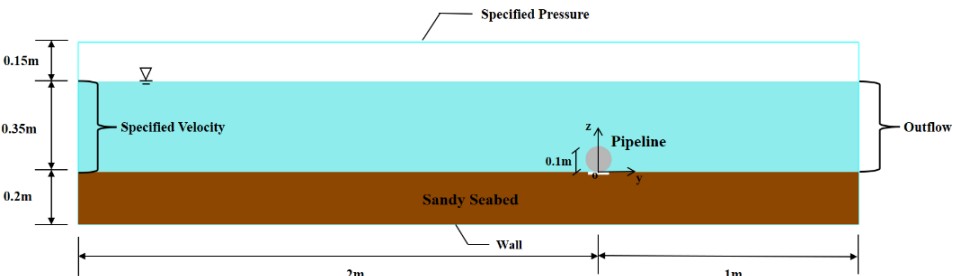

**Figure 1.** Sketch of numerical model setup.

A mesh sensitivity analysis was carried out using four meshes, i.e., Mesh A, B, C and D, as listed in Table 2. The numerical results of the 2D scour hole profile below the pipeline at the final stage of $t$ = 370 min are compared with the experimental result of Mao [2], as shown in Figure 3. The convergence of the simulation results is achieved at Mesh A and Mesh B. However, the total number of Mesh A is over three times that of Mech B. For Mesh B, the total cell number is 28,800, with a uniform grid size of 0.01 m in the *y*-direction and a minimum size of 0.002 m and a maximum size of 0.01 m in the *z*-direction. The minimum time step is set to $10^{-3}$ s. Considering both efficiency and accuracy, the mesh density level of Mesh B is adopted in the following studies.

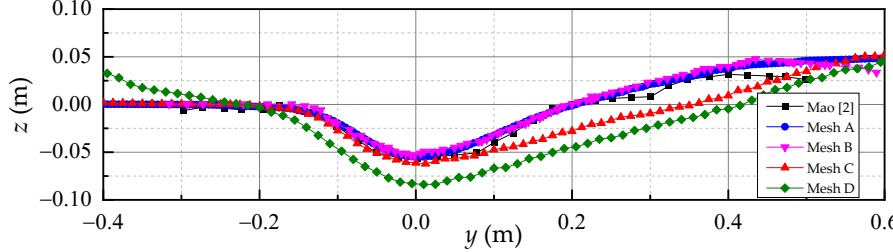

**Figure 2.** The numerical results of scour hole profile with different mesh levels ($t$ = 370 min).

**Table 2.** Cases for mesh sensitivity analysis.

| Case | Total Number of Cells | Y Direction | | | Z Direction | | |
|------|------------------------|-------------|-------------|-------------|-------------|-------------|-------------|
| | | Minimum Size (m) | Maximum Size (m) | Grid Number | Minimum Size (m) | Maximum Size (m) | Grid Number |
| Mesh A | 96,000 | 0.005 | 0.005 | 600 | 0.001 | 0.005 | 160 |
| Mesh B | 28,800 | 0.010 | 0.010 | 300 | 0.002 | 0.010 | 96 |
| Mesh C | 18,480 | 0.013 | 0.012 | 240 | 0.003 | 0.013 | 77 |
| Mesh D | 12,800 | 0.015 | 0.015 | 200 | 0.003 | 0.015 | 64 |

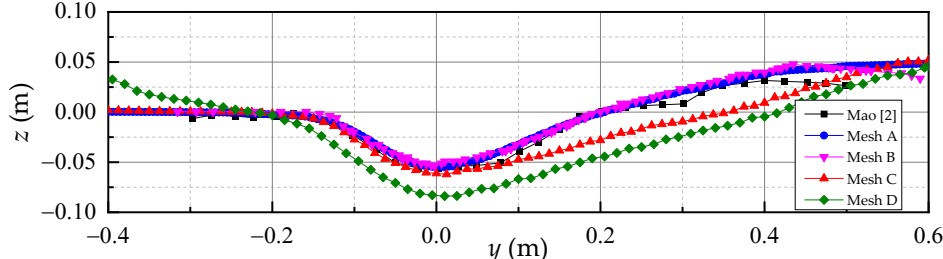

**Figure 3.** The numerical results of scour hole profile with different mesh levels (*t* = 370 min).

Figure 4 compares the present numerical results with the experimental result of Mao [2] and the numerical results of Liang et al. [10] and Xu [23] at three stages of scour process. At the early stage of *t* = 10 min, there is a rising sand dune downstream of the pipeline. The upstream slope of the sand dune is close to the pipeline, and the downstream slope is relatively steeper. The sand dune moves downstream gradually and disappears after *t* = 200 min, meanwhile, the scour depth below the pipeline increases slowly. The increasing and decreasing phases of the sand dune height correspond to the jet scour and the lee-wake scour, respectively, as reported by Mao [2]. The present numerical results are in good agreement with the experimental result of Mao [2] in terms of the position and maximum depth of the scour hole. Especially at *t* = 200 min, the scour profile in the present study is closer to the experimental result of Mao [2] than those of Liang et al. [10] and Xu [23].

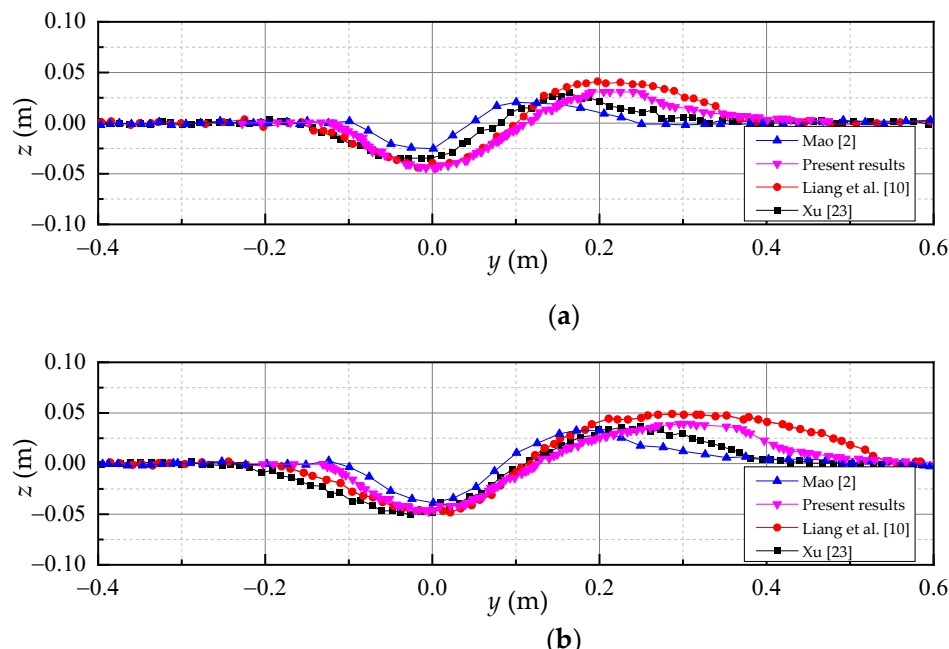

**Figure 4.** *Cont.*

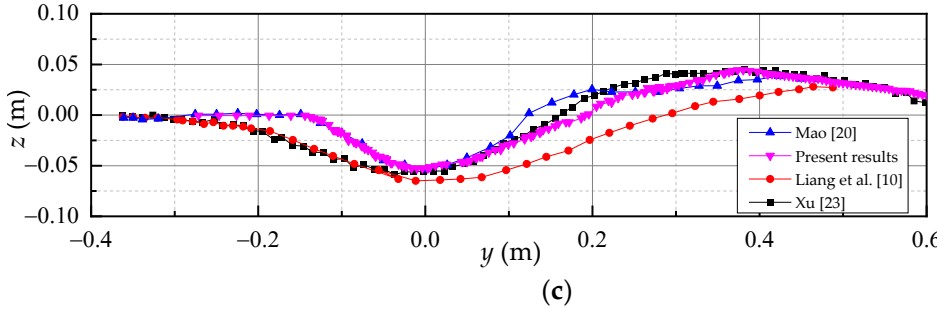

**(c)**

**Figure 4.** Comparison of scour hole profiles below the pipeline: (**a**) $t$ = 10 min; (**b**) $t$ = 30 min; (**c**) $t$ = 200 min.

### 2.2.2. Simulation of 3D Scour Process

In this section, the 3D scour propagation along the spanwise direction of a pipeline is simulated to further validate the numerical model. The flume width is 1.0 m. A pipeline with a diameter of 0.1 m is laid on the seabed with a zero-embedment depth. The related 3D scour model parameters are listed in Table 3. The current is perpendicular to the pipeline. To initialize the scour, an initial scour hole of 0.1 D was set below the center point of the pipeline, as implemented by Cheng et al. [16]. Then the 3D scour propagates along the span from the middle to the ends of the pipeline. Then, the numerical results are compared with those of Xu [23] and the empirical values of Cheng et al. [16].

**Table 3.** 3D-single pipeline scour model parameters.

| $v_o$ (m/s) | $D$ (m) | $p$ (kg/m³) | $d_{50}$ (mm) | $\theta$ | $e/D$ | $V_h$ (mm/s) | | |
|---|---|---|---|---|---|---|---|---|
| | | | | | | Present | Xu [23] | Cheng et al. [16] |
| 0.35 | 0.1 | 2650 | 0.36 | 0.040 | 0 | 0.811 | 0.793 | 0.802 |

Figure 5a shows the vertical profiles along the pipeline axis at four-time instances. The spanwise scour hole is basically symmetrical about the middle point when the pipeline is perpendicular to the current. Figure 5b shows the development of the free span length with time. The numerical result of Xu [23] is also plotted for comparison. The free span length increases almost linearly with time. It is observed that the present numerical result agrees well with that of Xu [23]. Then the propagation rate of the 3D scour along the pipeline ($V_h$) is calculated through the curves and is presented in Table 3. The present numerical result of $V_h$ agrees well with those published by Xu [23] and Cheng et al. [16], as shown in Table 3.

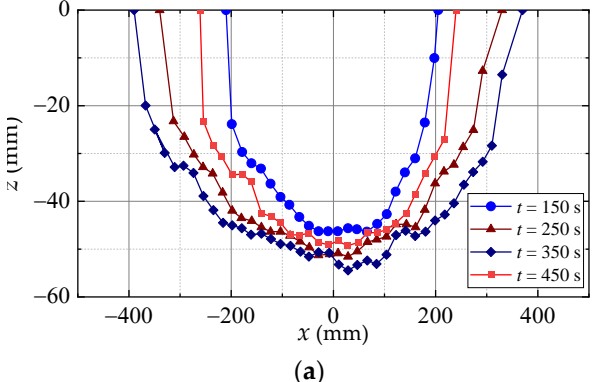

**(a)**

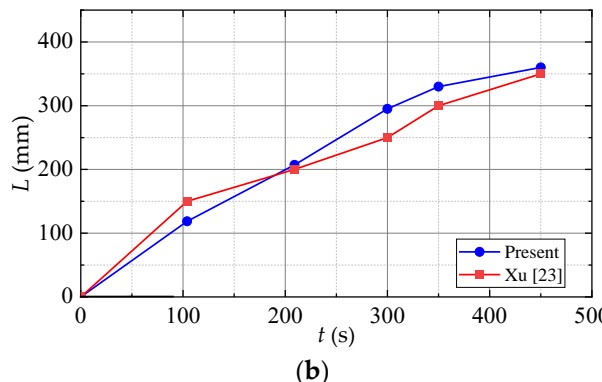

**(b)**

**Figure 5.** Numerical results of 3D scour below a pipeline, (**a**) vertical profiles along the pipeline axis, (**b**) development of span length with time.

The accuracy of the present scour model has been proved by the above verifications of the two- and three-dimensional scour processes. Thus, it can be applied to the 3D scour simulation of the two crossing pipelines in the next section.

## 3. Scour below Two Crossing Pipelines

A 3D numerical model for scour below two crossing pipelines is shown in Figure 6. The side view of the computation domain in the *y-z* plane is the same as that shown in Figure 4. As the development of scour depth below the crossing point of the pipelines is mainly focused rather than the 3D scour propagation, short segments of the pipeline are simulated in the present study. The flume has a width of 1.0 m, i.e., $x = -0.5$ m~0.5 m. The lower pipeline is fixed and perpendicular to the flow direction, laying directly on the seabed with a zero-embedment depth. The upper pipeline is placed above the lower one at their middle points with an intersecting angle $\alpha$. Here, $\alpha = 0°$ denotes that the two pipelines are parallel, and $\alpha = 90°$ denotes that the upper pipeline is placed along the flow direction and perpendicular to the lower one. The gap between two pipelines at the crossing point is set as zero. All flow and sediment parameters are the same as those listed in Table 3. Numerical simulations of scour process below the two pipelines are carried out at five intersecting angles, i.e., $\alpha = 0°$, 15°, 30°, 45° and 90°.

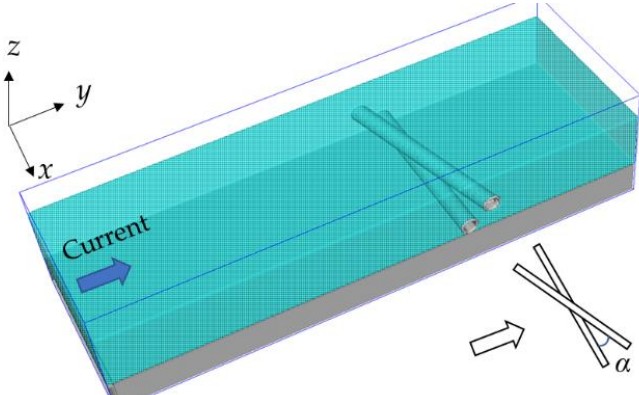

**Figure 6.** Sketch of the 3D numerical model for scour below two crossing pipelines.

Figure 7 shows the 3D seabed profiles below the two pipelines during scour process for $\alpha = 0°$. The shape of the scour hole is approximately symmetric with respect to the center point of the pipeline. The shape of the scour hole downstream of the pipeline is an isosceles triangle in the plane view, and the shape of the scour hole upstream of the pipeline is in an ellipse shape. The scour range upstream of the pipeline is larger than that downstream. It is observed that the scour hole propagates to the end of the pipeline at $t = 100$ s.

Figure 8 shows the 3D profiles of scour hole below the pipeline for $\alpha = 30°$. The scour hole in the spanwise direction shows asymmetry about the center point of the pipeline. With time elapsing, the asymmetry is becoming remarkable. The scour range on the left side is larger than that on the right side, especially downstream of the pipeline. In the present study, with the increase of the intersecting angle, the asymmetry of the scour hole is more significant than that for a small intersecting angle. It takes about 350 s for the scour hole to propagate to the end of the pipeline. Generally, with the increase of the intersecting angle, the time for scour propagation to the end of the pipeline is also larger than that of the smaller intersecting angle.

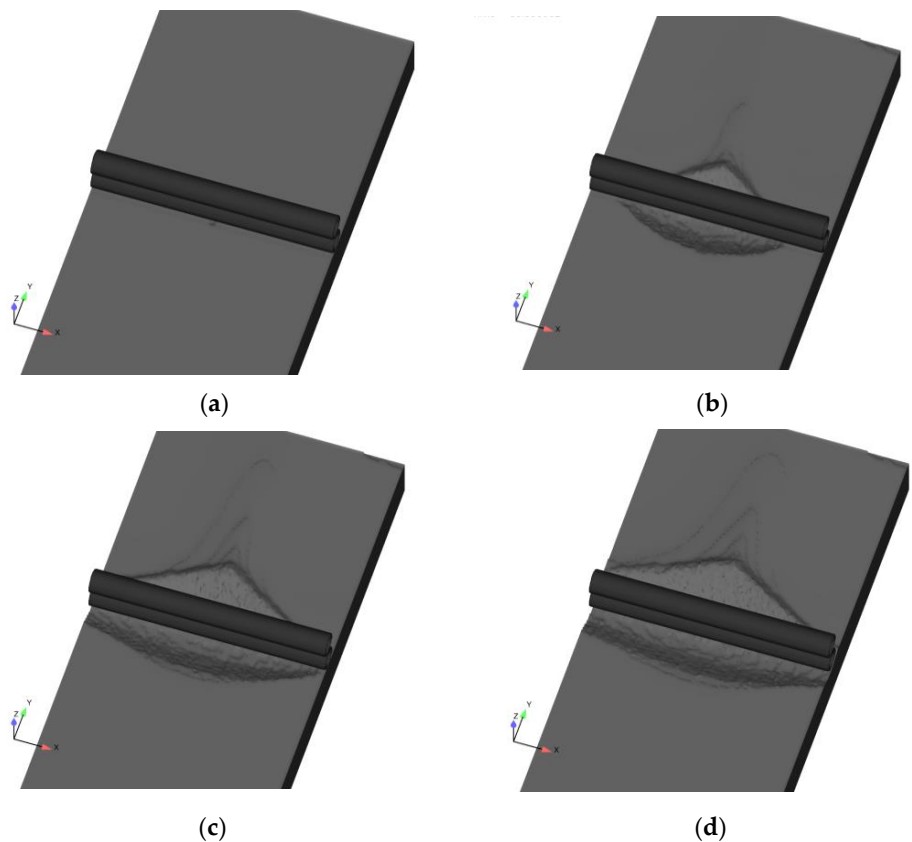

**Figure 7.** The 3D profiles of scour hole for $\alpha = 0°$: (**a**) $t = 0$ s; (**b**) $t = 50$ s; (**c**) $t = 100$ s; (**d**) $t = 125$ s.

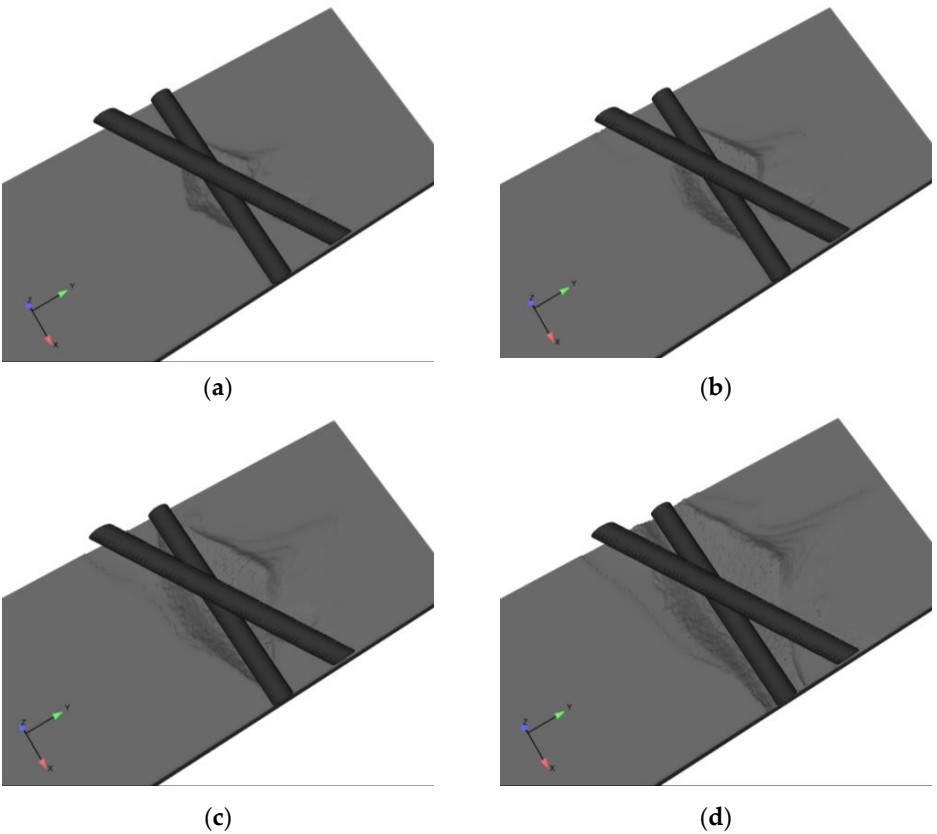

**Figure 8.** The 3D profiles of scour hole for $\alpha = 30°$: (**a**) $t = 0$ s; (**b**) $t = 100$ s; (**c**) $t = 200$ s; (**d**) $t = 350$ s.

Figure 9 compares the scour hole profiles at the cross section of the middle width of the flume (i.e., *x* = 0 m) for different intersecting angles at *t* = 300 s and 900 s. The scour hole profile for an isolated pipeline is also plotted for comparison. It can be found that due to the existence of the upper pipeline, the maximum scour depth and the width of scour hole below the lower pipeline are larger than those of an isolated pipeline, and the maximum scour depth and width of the scour hole decrease with the increase of the intersecting angle.

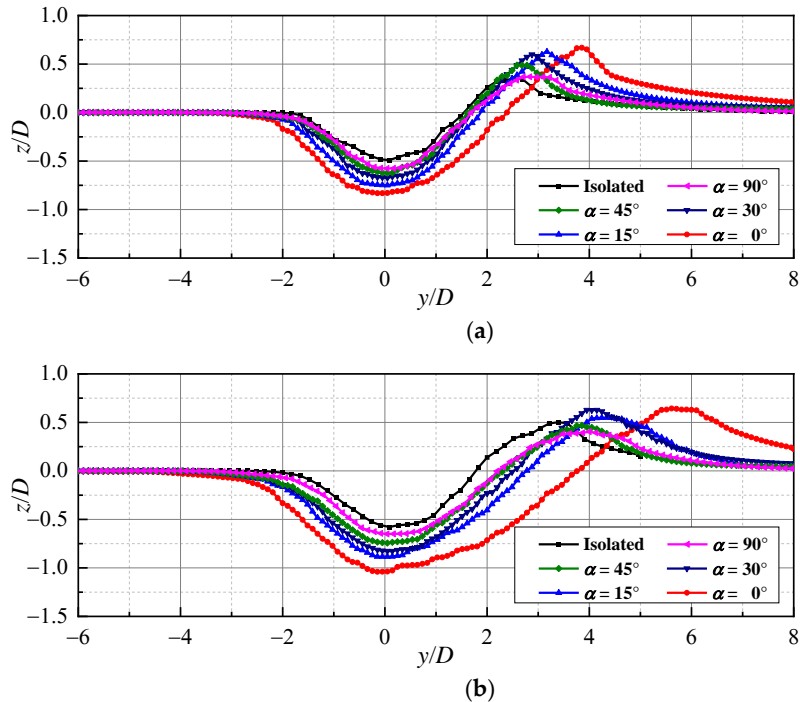

**Figure 9.** The scour profiles for different intersecting angles: (**a**) *t* = 300 s; (**b**) *t* = 900 s.

In the present study, the scour depth below the center point of the pipeline, i.e., the crossing position, is specifically investigated. Figure 10 shows the temporal development of the scour depth below the pipelines for *α* = 0, 15°, 30°, 45°, 90° and an isolated pipeline, respectively. The scour depth develops rapidly at the initial stage, then increases gently with time, finally reaching a final equilibrium state. The isolated pipeline has the minimum scour equilibrium depth in all cases, indicating that the crossing of the upper pipeline can enhance the equilibrium scour depth. The scour equilibrium depth *S* has the largest value for *α* = 0° and decreases with the increase of the intersecting angle. The scour equilibrium depth has the minimum value for *α* = 90°, which is still greater than that of an isolated pipeline.

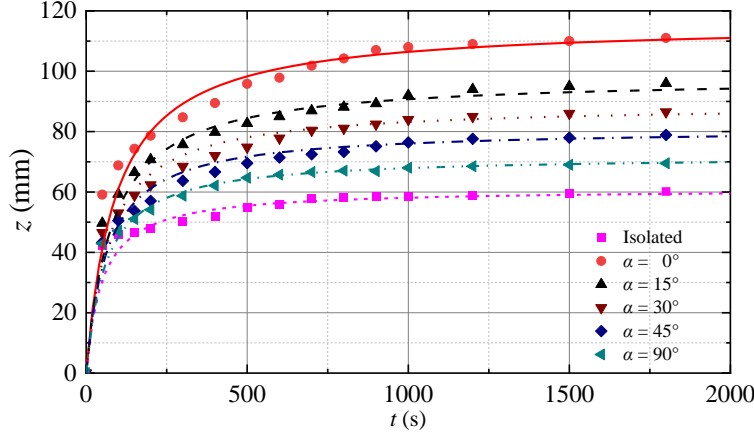

**Figure 10.** Temporal developments of scour depth for isolated and two crossing pipelines.

In studies of local scour below pipelines, the time scale of scour is also an important factor that needs to be assessed, in addition to the equilibrium scour depth. The time scale *T* is related to the rate at which scour develops and was first introduced to pipelines by Fredsøe et al. [3]. Its physical meaning is the duration of time necessary for the scour depth to reach a significant fraction of the equilibrium value. Based on the scour curve, the time scale of scour can be calculated with the integration methods as those adopted by Fuhrman et al. [15] and Zang et al. [8]. In the previous studies, the time scale is commonly investigated in the nondimensional form *T\**, which can be expressed as:

$$T^* = \frac{\left(g(s-1)d_{50}^3\right)^{1/2}}{D^2} T \tag{8}$$

Table 4 lists the equilibrium depth and time scale of scour below the crossing point of the pipelines for different intersecting angles. Their nondimensional forms, *S/D* and *T\**, are also calculated. Figure 11a shows the dimensionless equilibrium scour depth *S/D* varying with the intersecting angle $\alpha$. It is concluded that when another pipeline is laid above a pipeline with an intersecting angle, the equilibrium scour depth at the lower pipeline is always larger than the isolated case. The equilibrium scour depth for $\alpha = 0°$ has the maximum value and decreases with the increase of the intersecting angle. Therefore, if the crossing of pipelines is inevitable, the intersection angle shall be designed as large as possible so that the equilibrium scour depth can obtain a smaller value.

**Table 4.** Equilibrium depth and time scale of scour for different intersecting angles.

| Cases | S (mm) | S/D | T (s) | T* |
|---|---|---|---|---|
| Isolated | 60.2 | 0.60 | 155.5 | 0.43 |
| $\alpha = 90°$ | 69.5 | 0.70 | 157.8 | 0.43 |
| $\alpha = 45°$ | 78.9 | 0.79 | 180.7 | 0.50 |
| $\alpha = 30°$ | 86.5 | 0.87 | 192.3 | 0.53 |
| $\alpha = 15°$ | 96.3 | 0.96 | 202.2 | 0.56 |
| $\alpha = 0°$ | 111.3 | 1.11 | 207.4 | 0.57 |

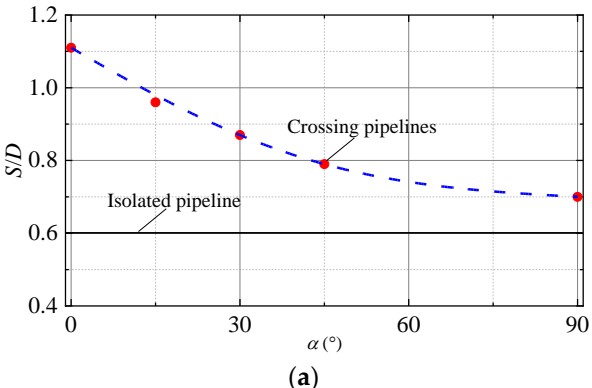
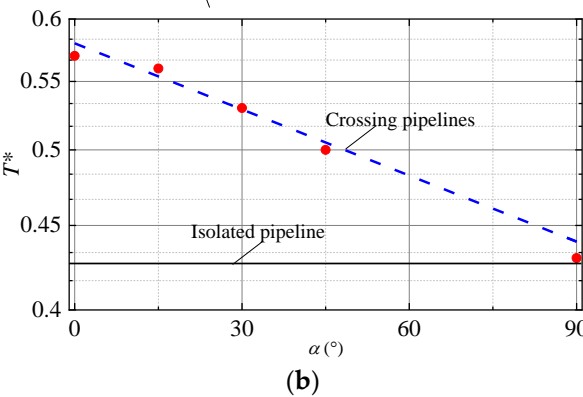

(**a**)       (**b**)

**Figure 11.** Variations of (**a**) *S/D* and (**b**) *T\** with intersecting angle $\alpha$.

Figure 11b shows the variation of the nondimensional time scale of scour with the intersecting angle. Generally, the nondimensional time scale for crossing pipelines decreases with the increase of the intersecting angle and is also larger than that for an isolated pipeline. The two parallel pipelines need the most time to reach the equilibrium status than any other intersecting angles because they also have the largest equilibrium scour depth.

## 4. Flow Structures around the Pipelines

Figure 12 shows the streamlines around the two parallel pipelines, i.e., $\alpha = 0°$, at different stages of scour process. At the initial stage of $t = 5$ s, a large vertical recirculation zone is formed downstream of the pipeline at each end. This is because the blockage effect induced by the two parallel pipelines is significant, which causes a large pressure difference between the upstream and downstream sides of the pipeline. Meanwhile, in the middle of the pipeline, the flow streamlines converge into the initial scour hole and form a pair of horizontal vortices downstream of the pipelines. The convergence of the streamlines in the scour hole increases the local bed shear stress, inducing the rapid development of scour depth at the initial stage. The pair of horizontal vortices downstream of the pipeline also enhance the spanwise propagation of the scour hole. With the development of the pipeline span, at $t = 25$ s, the streamlines keep converging into the scour hole, and the size of the horizontal vortices downstream of the scour hole is becoming large. Both results in the scour depth and free span developing at high rates. It is also observed that the vertical downstream recirculation zones at the end of the pipeline disappear due to the decrease in the pressure difference around the pipeline. At $t = 50$ s, the strength of the converging streamlines in the scour hole decreases, and the downstream vortices are also weak because the span of the pipeline reaches a sufficient length. The development rates of scour depth and free span decrease further. At $t = 150$ s, the scour hole propagates to both ends of the pipeline, the streamlines pass beneath the pipeline with no convergence, and the downstream vortices also disappear. The scour below the pipeline develops in a two-dimensional form. Generally, the flow fields are approximately symmetry about the middle point of the pipeline during the scour process.

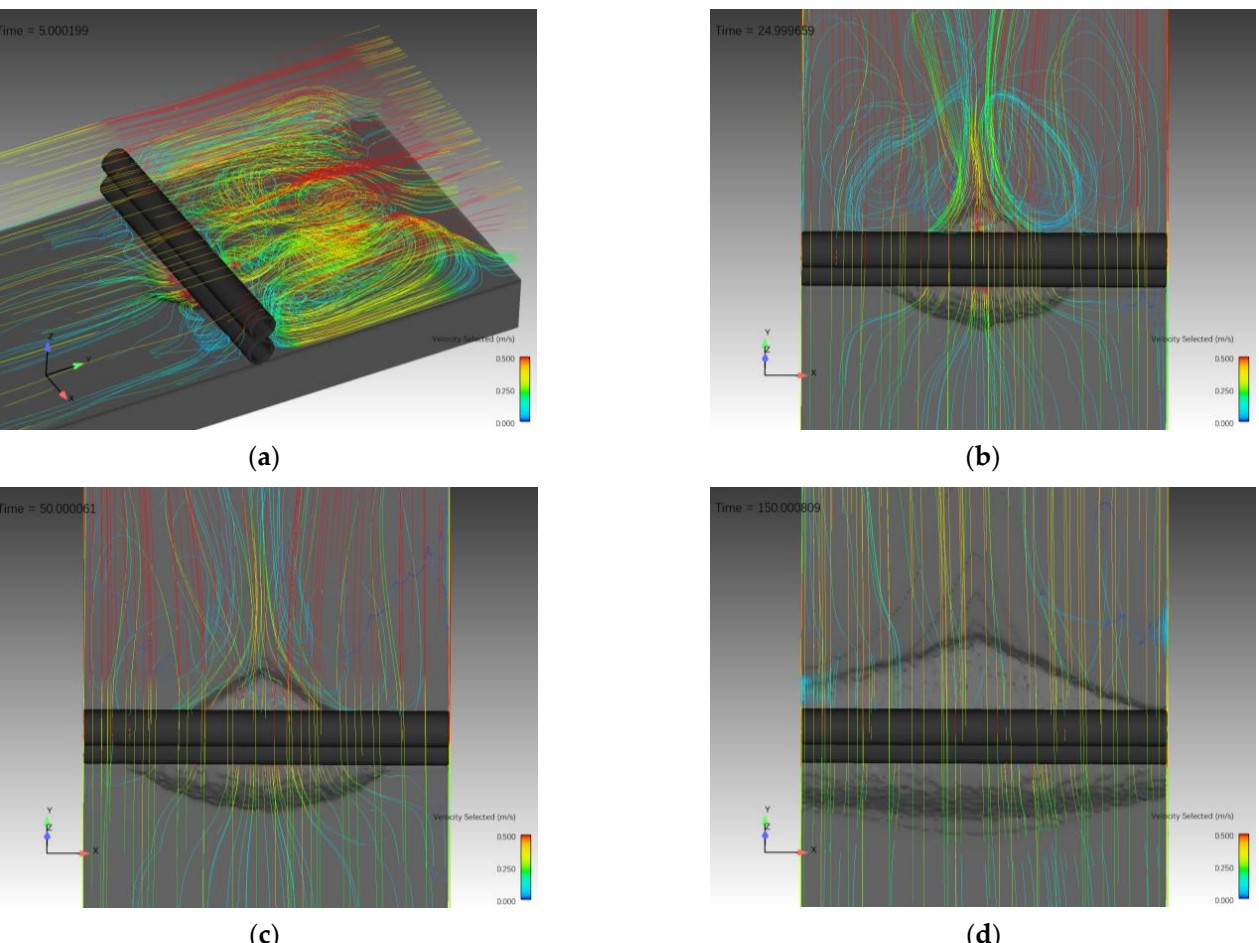

**Figure 12.** Streamlines around the pipelines for $\alpha = 0°$: (**a**) $t = 5$ s; (**b**) $t = 25$ s; (**c**) $t = 50$ s; (**d**) $t = 150$ s.

Figure 13 shows the flow structures around the two intersecting pipelines with an angle of 30° at different stages. It can be observed that the flow structures are more complex than those for the parallel case. At the initial stage of $t = 10$ s and 50 s, the streamlines converge in the scour hole and form strong vortices downstream of the scour hole. However, the downstream vortices are not symmetry about the middle point of the pipeline. The vortices mainly form at the left side of the scour hole, while the vortices at the right side are suppressed by the upper crossing pipeline. There is no recirculation zone formed downstream of the end of the pipeline. With the development of the free span, the vortices downstream of the scour hole disappear at $t = 200$ s. Meanwhile, a small vortex is formed at the right end of the lower pipeline. For $t = 900$ s, the free span propagates to both ends of the pipeline. The flow streamlines freely pass through the scour hole, and no vortex forms around the lower pipeline. Meanwhile, there is a spiral vortex forming along the axis of the upper pipeline from the center point extending to the downstream end. Generally, the scour propagation rate for $\alpha = 30°$ is much lower than that for $\alpha = 0°$.

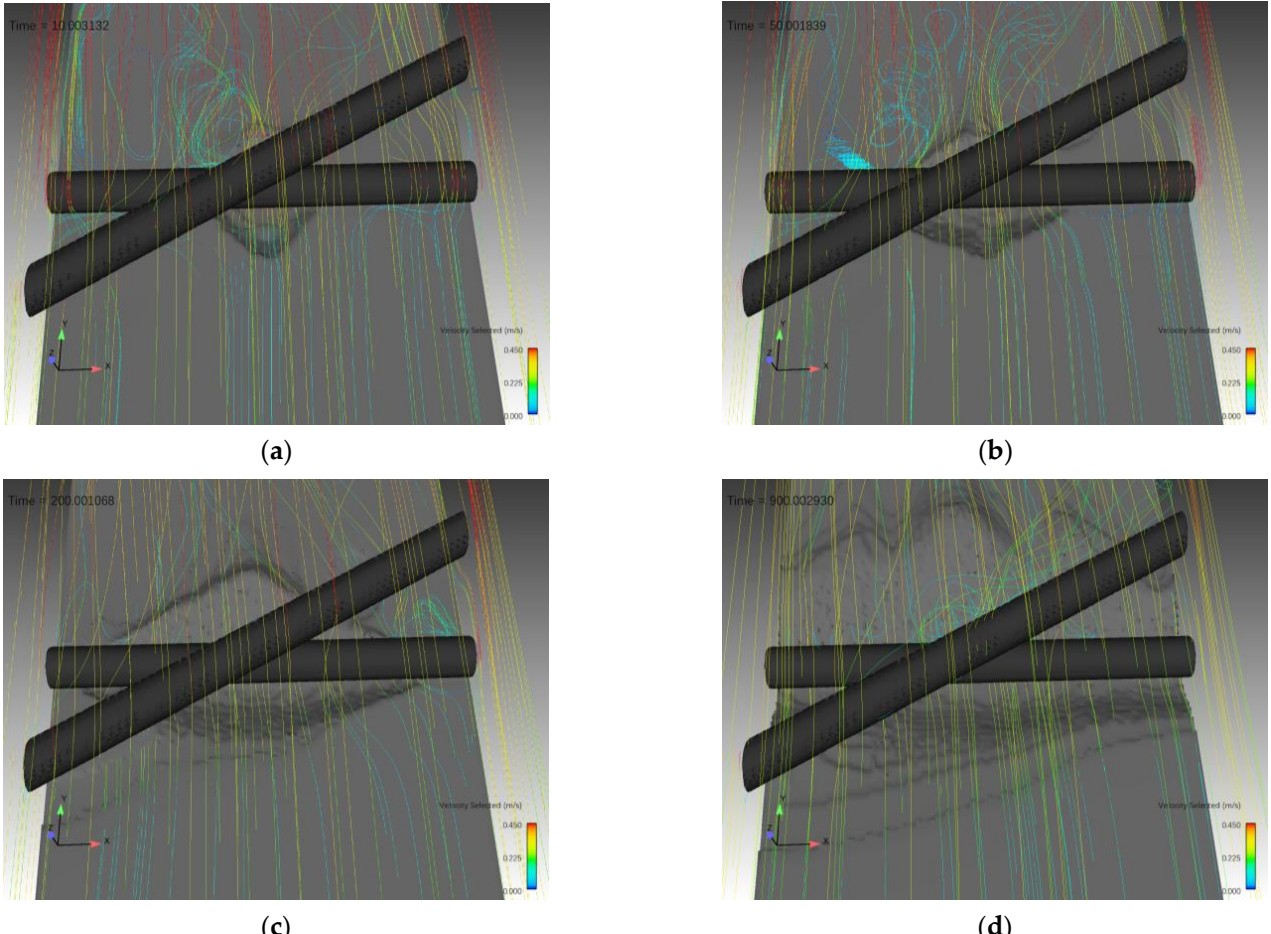

**Figure 13.** Streamlines around the pipelines for $\alpha = 30°$: (**a**) $t = 10$ s; (**b**) $t = 50$ s; (**c**) $t = 200$ s; (**d**) $t = 900$ s.

Due to the overlap of the upper pipeline on the existing lower pipeline, the flow near the seabed was blocked by the two pipelines, which enhances the local bed shear stress. This blockage effect is most significant for the two parallel pipelines ($\alpha = 0°$), while for the orthogonal case ($\alpha = 90°$), the blockage effect is weakest. The flow structures around the pipelines also indicate that the streamlines beneath the pipelines are densest for the parallel case, especially at the initial stage of the scour process, which accelerates the erosion rate. With the increase of the intersecting angle, the blockage effect weakens, which results in a smaller scour depth than that of the parallel case.

## 5. Concluding Remarks

In this paper, the CFD software Flow-3D is used to simulate the local scour of two crossing pipelines under steady flows. The present numerical model was validated first based on the simulation of the 2D and 3D scour process below a pipeline, and good agreements with the published results were obtained. Then, the effect of the intersecting angle on the equilibrium depth and time scale of scour below the pipeline is investigated. Here, the lower pipeline is fixed and perpendicular to the flow direction, while the upper pipeline is laid directly above the lower one with an intersecting angle. Two pipelines cross with each other at their middle points. The 3D scour for an isolated pipeline perpendicular to the current is also simulated for comparison. The following conclusions can be drawn:

When two pipelines cross with each other at an angle, both the equilibrium depth and the time scale of scour process below the crossing point of the pipelines are greater than those of an isolated single pipeline. The equilibrium depth and time scale of scour decrease with the increase of the intersecting angle. The equilibrium depth and time scale of scour have the maximum values when the two pipelines are parallel, i.e., $\alpha = 0°$, while they have the minimum values when the upper pipeline is perpendicular to the lower one, i.e., $\alpha = 90°$. This is mainly because the blockage effect by the two pipelines enhances the local bed shear stress beneath the pipelines and enlarges the size of the downstream vortices. This blockage effect is most significant for the parallel arrangement ($\alpha = 0°$), while it is weakest when the two pipelines are perpendicular to each other. The 3D visualization of the flow streamlines around the pipelines allows for showing the enhancement of flow acceleration below the pipelines. When the two pipelines are in a parallel arrangement, the convergence of the streamlines and the downstream vortices is strongest. The convergence of the streamlines in the scour hole becomes weak with the increase of intersecting angle. In practical engineering, if a crossing of two pipelines is inevitable, the intersecting angle of the two pipelines should be increased as large as possible to avoid excessive scour.

**Author Contributions:** Conceptualization, Z.Z. and M.Z.; methodology, F.Z., Z.Z. and M.Z.; investigation, Z.Z. and F.Z.; resources, Z.Z. and J.Z.; writing—original draft preparation, F.Z. and Z.Z.; writing—review and editing, Z.Z., M.Z. and J.Z.; visualization, F.Z.; supervision, M.Z. and B.X.; funding acquisition, Z.Z., B.X. and X.Z. All authors have read and agreed to the published version of the manuscript.

**Funding:** This research was funded by the National Natural Science Foundation of China (51579232, 51890913, 5197090657) and the Open Funding of State Key Laboratory of Hydraulic Engineering Simulation and Safety (HESS-2016).

**Institutional Review Board Statement:** Not applicable.

**Informed Consent Statement:** Not applicable.

**Data Availability Statement:** Data from the present experiment appear in the submitted manuscript.

**Conflicts of Interest:** The authors declare no conflict of interest.

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
