# Peer review of "Numerical Investigations on Scour and Flow around Two Crossing Pipelines on a Sandy Seabed"

_jmse, doi:10.3390/jmse10122019_

Round 1

Reviewer 1 Report

I found this article interesting and appropriate for publishing in JMSE. Some comments and suggestions are made in the attached file.

Reviewer 2 Report

Numerical Investigations on Scour and Flow around Two Intersecting Pipelines on a Sandy Seabedby Zhipeng Zang et al. This paper studies the blocking effect of submarine pipelines on ocean current through numerical simulation, and discusses the best pipeline crossing mode. In general, the research process of this paper is clear, the experimental setup is reasonable, and the discussion is sufficient. The conclusions obtained have a certain reference role for the actual project implementation. It is an academic paper of high quality. The following are my comments on this paper.

1.      Line 61, "SGS model", please give the full name in the text first, and then use the abbreviation.

2.      88 to 90 lines, the author should list references.

3.      154 lines have reference errors.

4.      123 lines, formula number is missing.

5.      In Figure 5, α The unit of is not displayed;

6.      In line 358, "Meanwhile, a small cortex is formed at the right end of the lower pipeline." The reviewer did not understand the meaning of this sentence;

7.      'The equilibrium sour depth' in line 403, 405 and 406 should be 'The equilibrium sour depth';
